# Challenging Sarcopenia: Exploring AdipoRon in Aging Skeletal Muscle as a Healthspan-Extending Shield

**DOI:** 10.3390/antiox13091073

**Published:** 2024-09-03

**Authors:** Camille M. Selvais, Maria A. Davis-López de Carrizosa, Romain Versele, Nicolas Dubuisson, Laurence Noel, Sonia M. Brichard, Michel Abou-Samra

**Affiliations:** 1Endocrinology, Diabetes and Nutrition Unit, Institute of Experimental and Clinical Research, UCLouvain, 1200 Brussels, Belgium; camille.selvais@uclouvain.be (C.M.S.); mayadavis@us.es (M.A.D.-L.d.C.); romain.versele@uclouvain.be (R.V.); nicolas.j.dubuisson@uclouvain.be (N.D.); laurence.noel@uclouvain.be (L.N.); sonia.brichard@uclouvain.be (S.M.B.); 2Departamento de Fisiología, Facultad de Biología, Universidad de Sevilla, 41004 Seville, Spain

**Keywords:** adiponectin, adiponectin agonist, sarcopenia, tubular aggregates, abnormal mitochondria, AMPK, autophagy

## Abstract

Sarcopenia, characterized by loss of muscle mass, quality, and function, poses significant risks in aging. We previously demonstrated that long-term treatment with AdipoRon (AR), an adiponectin receptor agonist, alleviated myosteatosis and muscle degeneration in middle-aged obese mice. This study aimed to determine if a shorter AR treatment could effectively offset sarcopenia in older mice. Two groups of old mice (20–23 months) were studied, one untreated (O) and one orally-treated with AR (O-AR) at 50 mg/kg/day for three months, compared with control 3-month-old young mice (Y) or 10-month-old young-adult mice (C-10). Results showed that AR remarkably inversed the loss of muscle mass by restoring the sarcopenia index and fiber count, which were greatly diminished with age. Additionally, AR successfully saved muscle quality of O mice by halving the accumulation of tubular aggregates and aberrant mitochondria, through AMPK pathway activation and enhanced autophagy. AR also bolstered muscle function by rescuing mitochondrial activity and improving exercise endurance. Finally, AR markedly curbed muscle fibrosis and mitigated local/systemic inflammation. Thus, a late three-month AR treatment successfully opposed sarcopenia and counteracted various hallmarks of aging, suggesting AR as a promising anti-aging therapy for skeletal muscles, potentially extending healthspan.

## 1. Introduction

The 30-year increase in life expectancy at birth over the last 100 years is one of humanity’s greatest achievements. Efforts are now focused on addressing aging-related diseases in order to both extend and improve the healthspan. Sarcopenia occurs commonly as an age-related process. It is a generalized skeletal muscle disorder characterized by the accelerated loss of muscle mass, quality, and function. Sarcopenia is associated with adverse outcomes such as falls, frailty, institutionalization, and mortality. Physical activity is currently the first therapeutic strategy to prevent sarcopenia but is not always feasible, and no specific drug has yet been approved [1]. Preventive/therapeutic pharmacological approaches are therefore needed.

Adiponectin (ApN) is a hormone primarily secreted by adipocytes, exerting insulin-sensitizing, fat-burning, and anti-inflammatory effects in various tissues, including the skeletal muscle. AdipoR1 and AdipoR2 are the major ApN receptors in vivo, with AdipoR1 being the muscle main receptor [2]. Regarding skeletal muscle, we have previously demonstrated that ApN exhibits potent protective effects in mdx mice, a model of Duchenne Muscular Dystrophy (DMD), a severe muscle disease. Indeed, ApN reduces muscle inflammation and oxidative stress and enhances the myogenic program, thereby decreasing muscle damage while increasing force/endurance. This protection is mainly mediated through the AdipoR1-AMP-activated protein kinase (AMPK)-peroxisome proliferator-activated receptor coactivator-1α (PGC-1α) pathway [2,3]. Importantly, ApN overexpression in transgenic mice prolongs lifespan and healthspan [4,5], whereas ApN deficiency in knockout mice shortens them [5]. Thus, these results support a role of ApN as potent protector of skeletal muscle and an essential regulator of healthspan and lifespan.

ApN possesses a complex three-dimensional structure, making its production and administration challenging. It typically must be injected to exert its effects [2]. Hence, the development of novel molecules that replicate ApN’s benefits and can be easily administered is pertinent. AdipoRon, an orally active synthetic agonist of ApN receptors, has shown promise in animal models and has been proposed for the treatment of type 2 diabetes and other obesity-related disorders [2,6]. Relevant for translational applications, AdipoRon retains efficacy in AdipoR1-humanized mice [7]. Our research has revealed that, like ApN, AdipoRon protects the skeletal muscle of mdx mice [8]. So far, its potential benefits on muscle aging and wasting have been scarcely addressed. Two new studies highlighted the potent anti-inflammatory action of AdipoRon on skeletal muscle, in mouse models of sarcopenic obesity [9] or cancer cachexia [10]. Additionally, we recently found that long-term AdipoRon treatment (1-yr) significantly improved endurance in middle-aged obese mice and remarkably reduced myosteatosis and muscle degenerative markers, promoting healthy aging [11]. Finally, short-term (6-wks) i.v. administration of AdipoRon has been tested in older non-obese mice. This treatment was found to improve muscle function, accompanied by a slight increase in mitochondrial function. However, since there was no young control group, it remains unclear whether these old mice were truly sarcopenic. Moreover, conclusive evidence regarding AdipoRon’s ability to reverse potential muscle mass loss was lacking (no muscle weight provided and decreased size for almost all fiber types) [12]. Therefore, studies unambiguously demonstrating positive effects of AdipoRon on sarcopenia are warranted.

This study aimed to determine if a late AdipoRon treatment could effectively reverse sarcopenia and hallmarks of aging in older mice. To this end, old mice were orally administered AdipoRon for 3 months, and their muscle mass, quality, function, and circulating pro-inflammatory parameters were thoroughly assessed. Furthermore, the mechanisms underlying AdipoRon’s protective effects were extensively investigated, significantly broadening the scope of previous studies [11,12].

## 2. Materials and Methods

### 2.1. Animals

Male C57BL/6J mice aged 19.5 months were purchased from The Jackson Laboratory (Bar Harbor, ME, USA). Mice were first acclimatized and monitored for two weeks. At 20 months of age, old mice were matched for body weight and food intake, then divided into two groups and studied until 23 months of age: one group was left untreated (O, old mice, n = 10) and one was orally treated with AdipoRon (O-AR, n = 9) for 3 months. During the study, 3 mice from the untreated O group developed large intra-abdominal tumors, while this was the case for only one in the treated O-AR group. All these mice were excluded from data analyses. Mice were fed a standard diet (Carfil quality, Oud-Turnhout, Belgium). Throughout the protocol, drinking water was replaced by solidified water for all mice (Solid Drink, Triple A Trading, Tiel, The Netherlands) and renewed daily. In the treated group (O-AR), AdipoRon (Bio-Techne, Minneapolis, MN, USA) was incorporated into the solid drink at a dose of 50 mg/kg/day for a total duration of 3 months. Mice were compared with a group of 3-month-old young mice (Y, n = 6). Notably, 3-month-old mice are equivalent to young 20 yr-old humans [13] and are routinely used as controls for sarcopenic mice [14,15]. Another group of 10-month-old mice, whose body weight was similar to O mice [16], served as an additional control group (C-10, n = 10) for muscle morphometric analysis only.

Animals were maintained under the same standard-rearing conditions, i.e., constant temperature (22 °C) and humidity (40–60%), with a fixed 12 h light–12 h dark cycle (lights on from 7 a.m. to 7 p.m.), in environmental enriched cages, with social housing and cleaning of the microenvironment once a week. Each animal body weight and food consumption were closely monitored and measured once a week. Fasting blood glucose and insulinemia were measured at 22 months of age and an insulin resistance index was calculated as Insulin (ng/mL) × Glycemia (mg/dL) [17]. A functional treadmill test was performed at 22.5 months [11].

Y-mice were sacrificed at 3 months of age and O-mice were euthanized at 23 months of age, as described [8,11]. Blood samples were collected after sacrifice and stored at −20 °C for later analyses. Different skeletal muscles from the upper and lower limbs were collected and weighed, as well as liver, heart, subcutaneous, epididymal, brown adipose tissues, and tibia. Tibial length was measured using a caliper. Organs and tissues were either directly processed and studied in some experiments, or were stored at −80 °C for subsequent analyses. Electron microscopy and immunofluorescence analyses were performed on 3 and 4–6 mice/group, respectively.

### 2.2. In Vivo Functional Testing

Mice were subjected to a treadmill exhaustion test to assess their endurance and their resistance to fatigue. In this test, mice ran to exhaustion on a treadmill. An electric grid, placed at the back of the treadmill, delivered small electric shocks of a constant intensity (0.2 mA) to motivate the mice to run. The day before testing, mice were trained to run for 10 min at a speed of 3 m/min, which then gradually increased to reach a speed of 12 m/min. A treadmill exhaustion test was performed at 22.5 months of age in old mice and at 10 weeks of age in Y ones, where mice ran on a treadmill with a 5° upward incline. The initial speed was set at 5 m/min for the first 5 min and 10 m/min for the next 5 min. The speed was then progressively increased by 1.2 m/min to reach a maximum speed of 22.8 m/min. The maximum distance was set at 1200 m.

### 2.3. Quantification of Circulating Parameters

Blood glucose was measured using a glucometer (Freestyle Precision Neo, Abbott, Wavre, Belgium). ELISA kits were used to quantify Adiponectin (Abcam, Amsterdam, The Netherlands) and Insulin (Crystal Chem, Zaandam, The Netherlands) levels. Plasma cholesterol, triglycerides, and NEFAS were measured by commercial kits (all three from Diasys, Connecticut, CT, USA). These kits were based on colorimetric methods and were used following the manufacturers’ instructions. Multiplex assays were also designed to measure plasma concentrations of TNF-α, MCP-1, IFN-γ, IL-6, IL-10, MIP-1α, MIP-1β and MIP-2, (Meso Scale Discovery, Rockville, MD, USA).

### 2.4. Bright-Field Histochemistry and Morphometry

Tibialis anterior muscles were embedded in an optimal cutting temperature compound (OCT, Gentaur, Kampenhout, Belgium) and frozen in isopentane chilled with liquid nitrogen (VWR International, Leuven, Belgium). Then, 10 μm transversal cryosections were stained with Hematoxylin–Eosin. In addition, Gomori Trichrome (Kyvobio, Evere, Belgium) and Picrosirius red (Abcam) staining were used to quantify tubular aggregates and fibrosis, respectively. Finally, COX staining was performed to evaluate mitochondrial respiratory function as previously described [18]. For each staining, all slides were processed simultaneously. Whole muscle sections were scanned using a slide scanner (Pannoramic Scan II 3DHISTECH, Budapest, Hungary), and quantification was performed using Fiji/ImageJ 1.53k software (NIH, Bethesda, MD, USA) [19]. To quantify tubular aggregates, the percentage of the cross-sectional area covered with bright pink abnormal Gomori Trichrome inclusions was calculated. Fibrotic tissue was quantified after setting a color threshold and data obtained were expressed as the percentage of total section area. For analysis of COX, the images were converted to grayscale (8-bit, 0–255). By setting 3 different thresholds, the percentage of muscle area covered by 3 different intensities of staining (pale, intermediate, or dark) was quantified in intracellular regions.

### 2.5. (Immuno) Fluorescence—Fiber Typing

Cryosections of tibialis anterior were obtained as described above. Muscle sections were immunolabeled for the different myosin heavy chains (MyHCs), as described [20]. Briefly, mouse primary antibodies were directed against MyHC-1 (IgG2b and BA-D5), MyHC-2a (IgG1 and SC-71), MyHC-2x (IgM and 6H1) (Developmental Studies Hybridoma Bank, University of Iowa, IA), and rat primary antibody against laminin α-2 chain (Sigma-Aldrich, Overijse, Belgium) (1:10, 1:10, 1:5, and 1:1000, respectively). The corresponding secondary antibodies used were goat anti-mouse IgG2b Alexa Fluor (AF) 405 (1:500, Sigma-Aldrich), goat anti-mouse IgG1 AF488 (1:500, Thermo Fisher Scientific, Merelbeke, Belgium), goat anti-mouse IgM AF568 (1:1000, Abcam), and goat anti-rat AF647 (1:500, Thermo Fisher Scientific), respectively. Images were captured using an epifluorescence microscope (Axio Imager Z1/ApoTome1, Zeiss, Zaventem, Belgium). Excitation filters for DAPI (405 nm), FITC, TRITC, and Cy5 were selected to detect type 1, 2a, 2x fibers, and laminin, respectively. Type 2b fibers were recognized as unlabeled fibers (black sarcoplasm surrounded by laminin). Analysis of the fiber number and type was performed using the Fiji/ImageJ software. Results are presented as the total number of fibers over an entire cross-section and the number of fibers of each type. The minimum Feret’s diameter was calculated from laminin staining.

### 2.6. Transmission Electron Microscopy

Electron microscopy analysis was conducted on the gastrocnemius muscles of 3 mice per group. This number of mice is commonly employed in TEM-based studies to investigate mitochondrial morphology and ultrastructure [21]. The red and oxidative part of the gastrocnemius muscle was cut and fixed for 2 h in a solution of 2.5% glutaraldehyde in 0.1 M cacodylate buffer (pH 7.4). Samples were post-fixed for 1 h at 4 °C in 1% osmium tetroxide and then incubated with 2% uranyl acetate for 2 h at RT. Muscles were embedded in resin (Araldite) and transverse ultrathin sections were cut (70 nm thick) with an ultramicrotome (UC7, Leica Biosystems) and mounted on copper grids to be visualized with the MET LIBRA 120 (Zeiss).

Individual subsarcolemmal (SS) mitochondria from 3 Y, 3 O, and 3 O-AR mice were manually traced on TEM cross-sectional images using the Fiji/ImageJ software to quantify the mitochondrion area (μm^2^). On each image, the SS region was also traced and its area measured. The mitochondrial density was calculated by dividing the number of mitochondria by the SS area (N mitochondria/µm^2^ ratio). Around 45 images per group were analyzed, equivalent to 1117, 1604, and 1435 SS mitochondria sampled for Y, O, and O-AR, respectively. Mitochondria were classified as either “healthy” or “damaged”. Damaged mitochondria showed at least one of the following characteristics: disarticulated cristae and vacuolization, reduced electron density of the matrix, or swollen appearance [22,23]. Mitochondria with an area greater than 0.78 µm^2^ were considered swollen (represent the 90th percentile for the mitochondrial area of Y mice) [24]. The percentage of abnormal mitochondria, calculated as the number of damaged mitochondria out of the total number of mitochondria, was also measured.

### 2.7. RNA Extraction and Real-Time Quantitative PCR (RT-qPCR)

RNA was isolated from rectus femoris (quadriceps muscles) with TriPure reagent (Sigma-Aldrich). In total, 1 µg of total RNA was reverse transcribed, and 40 ng total RNA equivalents were amplified using an iCycler iQ real-time PCR detection system (Bio-Rad Laboratories, Nazareth, Belgium) as described [8]. RT-qPCR primers for mouse TATA-binding protein were ordered from Integrated DNA Technologies (Coralville, IA, USA). RT-qPCR primers for Myosin heavy chain 7 (Myh7) were used as reported [25]. New mouse primers for p62 were sense, 5′-GAT AGC CTT GGA GTC GGT GGG-3′, and antisense, 3′-CCG GGG ATC AGC CTC TGT AG-5′. The threshold cycles (Ct) were measured in separate tubes and in duplicate. The identity and purity of the amplified product were checked by electrophoresis on agarose minigels and analysis of the melting curve was carried out at the end of the amplification. To ensure the quality of the measurements, each plate included a negative control for each set of primers.

### 2.8. Protein Extraction, ELISA, and Western Blot

Tibialis anterior and triceps brachii were used for protein analyses, the fiber type composition of these two muscles being highly similar [26,27]. Muscle samples were homogenized in a lysis buffer supplemented with a 1% protease/phosphatase inhibitor cocktail (both from Cell Signaling Technology, BIOKE, Leiden, The Netherlands) and 10 mM NaF (Merck Life Science, Overijse, Belgium). Proteins were quantified using the Bradford method and were stored at 80 °C. In total, 25–200 µg of total protein extracts were used for each analysis. Six ELISA assays were used to specifically detect and quantify the phosphorylated and active form of AMP-activated protein kinase alpha (P-AMPK and Thr172), mammalian target of rapamycin (P-mTOR and Ser2448), UNC-51 like kinase 1 (P-ULK1 and Ser555), beclin-1 (P-beclin-1 and Ser 30), SMAD Family Member 2 (P-SMAD2 and Ser 465/467), and the p65 subunit of nuclear factor kappa B (NF-κB) (P-p65) (all from Cell Signaling Technology). Other ELISA kits were also used to detect Atrogin-1, muscle-specific RING-finger protein 1 (MuRF1), adiponectin receptor 1 (AdipoR1), peroxisome proliferator-activated receptor gamma coactivator 1-alpha (PGC-1α), translocase of the outer mitochondrial membrane complex subunit 20 (TOMM20), p16, p21 (all from MyBioSource, Vancouver, BC, Canada), 4-Hydroxynonenal (HNE) (Abcam), and Myogenin (Antibodies Online, Atlanta, GA, USA). Kits were based on colorimetric methods and were carried out following manufacturer’s instructions. Immunoblotting was performed as reported [3] by using rabbit monoclonal antibodies directed against the ubiquitin-binding protein p62, LC3A-I/II, and Beclin-1 (all from Cell Signaling Technology) [28]. Signals were revealed by enhanced chemiluminescence, then quantified and normalized to those of Ponceau stain [29] using ImageJ.

### 2.9. Statistical Analyses

Results are expressed as means ± SEM for the indicated number of mice. Comparisons between the three groups of mice (Y, O, and O-AR) were carried out by one-way ANOVA followed by Tukey’s test (Prism 9; Graphpad Software version 10.2.1, San Diego, CA, USA). Differences were considered statistically significant at *p* < 0.05.

## 3. Results

### 3.1. AdipoRon Mitigates Insulin Resistance, Rescues Endurance, and Protects Muscle Mass

One group of 20-month-old mice was orally treated with AdipoRon (O-AR, n = 10) for 3 months, while another was left untreated (Figure 1a). During the course of the study, 3 out of 9 mice in the untreated O group developed large intra-abdominal tumors, whereas this was the case for only 1 out of 10 mice in the treated O-AR group (these mice were excluded from the study). Young 3-month-old mice (Y) were used as controls for all parameters studied.

At the end of the study, body weight and cumulative food intake of O mice were higher than Y mice (~30%). Treatment with AdipoRon did not affect these parameters (Figure 1a). The insulin resistance index derived from 6h-fasting glycemia and insulinemia (Appendix A) doubled in O mice compared to Y ones but decreased by 20% under AdipoRon (Figure 1b). Circulating lipid parameters as well as plasma ApN levels did not change between the three groups (Appendix A). Using a treadmill exhaustion test, we evaluated the muscle function of O mice at 22.5 months of age. Endurance capacity, expressed as the distance covered until exhaustion, was markedly reduced in O mice compared to Y ones, while it was partially rescued by AdipoRon (+60% vs. O mice) (Figure 1c).

Adipose tissue weights sampled from different sites did not change significantly between the three groups (Appendix A). On the contrary, weights from some hind-limb muscles decreased with age [tibialis anterior (TA), Figure 1d, and gastrocnemius (G), Appendix A)]. The sarcopenia index, calculated as the ratio of skeletal muscle weight to tibia length [30], was significantly reduced in TA of O mice but was restored in O-AR ones, suggesting a therapeutic protection of AdipoRon against sarcopenia (Figure 1d). Similar findings were even more strikingly observed for G muscle (Appendix A), while in soleus, the decrease observed in O vs. Y mice was no longer significant under AdipoRon. To microscopically assess AdipoRon’s impact on sarcopenia, we analyzed TA cryosections, conducting fiber typing via immunofluorescence staining for myosin heavy chain isoforms and laminin staining for fiber size. (Figure 1e). The total number of fibers was significantly decreased in O mice when compared to Y ones, but was rescued in the O-AR group (Figure 1e). As expected, the TA contained a large number of fast-glycolytic type 2b fibers (dark), followed by 2x fibers (red), and a smaller number of slow-oxidative type 2a fibers (green) (Figure 1e). The number of 2x fibers was decreased in both groups of old mice. However, for each type, O-AR mice tended to have a higher number of fibers than O mice. Although each trend did not reach significance, taken together, they are in line with the preserved total number of fibers observed in O-AR mice (Figure 1e). Moreover, levels of Myogenin, an essential regulator of adult myofiber growth [2,3], were reduced by 10% in untreated O mice compared with Y mice, while levels were corrected with treatment (Appendix A). Fiber size was assessed using the Minimum Feret Diameter. Overall, fiber size was consistent across all groups (Figure 1f). However, O-AR mice showed a higher proportion of fibers in the middle range of the size distribution (Figure 1f). We also used an additional control group of adult mice (10-month-old; C-10) whose body weight (35.4 ± 1.0 g, n = 10) was similar to O mice (35.9 ± 2.2 g, n = 6) and muscle weight heavier than Y ones (*p* < 0.01 or less for TA, G, and S). When compared to C-10, O mice still exhibited a reduction in all sarcopenic indices (except EDL) (Appendix A) and fiber number in O mice (Appendix A), but further showed size atrophy specially at the expense of Type 2b (Appendix A). Accordingly, fiber size distribution was shifted to the left (Appendix A). Adiporon again rescued the sarcopenic indices (TA, G); fiber number but did not modify fiber size. Taken together, these results show that AdipoRon partly protects against age-related decline in muscle mass by preserving myofiber loss.

### 3.2. AdipoRon Reduces Accumulation of Tubular Aggregates and Promotes Autophagy

The accumulation of abnormal structures related to aging and identified as tubular aggregates (TAgs) was detected in transverse sections of TA muscle. TAgs appear as pale or slightly basophilic inclusions with hematoxylin–eosin staining (Figure 2a, upper panels, white arrows) and as bright pink ones with Gomori trichrome (lower panels, white arrows). By electron microscopy, TAgs form straight, single-, or double-walled tubules, regularly organized into tight aggregates with a honeycomb-like para-crystalline order (Figure 2c). These abnormal structures (bright pink inclusion) were quantified on Gomori sections from three groups of mice. They were absent in Y mice but abundant in O mice. Treatment with AdipoRon more than halved TAg accumulation compared to O mice (Figure 2b).

Since TAgs are a hallmark of dysregulated protein homeostasis, two key systems of protein clearance were studied: the ubiquitin–proteasome pathway and the autophagy–lysosome pathway. In the ubiquitin–proteasome system, ubiquitin ligases such as Atrogin-1 and Muscle-specific RING finger protein 1 (MURF1) link ubiquitin chains to proteins that are to be degraded by the proteasome. Both protein levels were decreased in O and O-AR mice (~40% vs. Y mice) (Figure 2d). For the autophagy–lysosome system, the microtubule-associated protein light chain 3 (LC3) II/I ratio and p62 protein levels were measured [28]. LC3-I is converted to LC3-II during the early stages of autophagy, and an increase in LC3-II/LC3-I ratio indicates enhanced autophagic activity. In the O-AR group, the ratio was increased by 50% (Figure 2e), mainly due to the upregulation of LC3-II (Appendix A). In contrast, p62 co-localizes with ubiquitinated protein aggregates and is efficiently degraded by autophagy. Thus, p62 accumulation inversely correlates with autophagic activity. p62 levels rose by 65% in O mice compared with Y mice, but reverted to Y values in O-AR mice (Figure 2e). Interestingly, p62 mRNA levels were increased in O-AR mice (Appendix A). The lack of p62 accumulation, despite enhanced transcription, further backs the idea of heightened degradation and thus increased autophagic flux under AdipoRon. These data suggest a downregulation of the ubiquitin–proteasome system with age, while AdipoRon enhances autophagy.

To unravel the mechanisms by which AdipoRon could increase autophagy, we studied the protein levels of the receptor and the activity of several downstream kinases known to be activated by ApN (Figure 3a) [2]. First, protein levels of AdipoR1 (ApN main muscle receptor) were decreased in O mice compared with Y ones (~35%), but were restored by AdipoRon treatment (Figure 3b). Second, the phosphorylated and active form of AMPK (P-AMPK), which tended to decrease in O mice, was overactivated and doubled in the O-AR group (Figure 3c). AMPK is the ApN main muscle signaling pathway [2] and has been recognized as an activator of autophagy through phosphorylation and activation of UNC-51 like kinase (ULK1) [31] and indirect inhibition of the mammalian target of rapamycin (mTOR) [32] (Figure 3a). Accordingly, the active phosphorylated form of mTOR was increased in both groups of O mice compared to Y ones, but to a lesser extent in O-AR mice (40% and 25%, respectively) (Figure 3d). It should be noted that chronic activation of mTOR in muscle may be catabolic, rather than anabolic [33]. AdipoRon also enhanced the phosphorylation and activation of ULK1 (30%) (Figure 3e). Eventually, both AMPK and ULK1 target Beclin-1 for autophagy activation (Figure 3a) [34]: the active phosphorylated form of Beclin-1 was doubled under AdipoRon treatment (Figure 3f and Appendix A). Taken together, these results show that AdipoRon activates autophagy in skeletal muscle, which could certainly contribute to its protective role against the abnormal accumulation of TAgs.

### 3.3. AdipoRon Enhances Mitochondrial Content, Function, and Normalizes Morphology

Mitochondrial content, function and morphology play a crucial role in muscle health. Notably, aging is associated with mitochondrial dysfunction, contributing to muscle decline [29]. PGC-1α, a transcriptional co-activator controlled by AMPK, is a master regulator of mitochondrial biogenesis and function (Figure 4a). Protein levels of PGC-1α were halved in O-mice when compared to Y ones, but significantly restored in O-AR mice (+36% vs. O mice) (Figure 4b).

Protein expression of the Translocase of Outer Mitochondrial Membrane 20 (TOMM20) was measured to assess muscle mitochondrial content (Figure 4a). TOMM20 was slightly but significantly increased in O-AR mice compared to the other two groups (12%), suggesting a higher mitochondria number (Figure 4c). Mitochondrial function was next studied by measuring Cytochrome C oxidase (COX) activity (Figure 4a,d). The proportion of dark fibers (i.e., with high activity of COX) was decreased in O mice compared to Y, indicating impaired mitochondrial activity with aging. By contrast, the proportion of dark fibers was 2-fold higher under AdipoRon and reached values observed in Y mice, suggesting a protective effect of the treatment (Figure 4d). In line with the stimulation of oxidative metabolism, AdipoRon doubled gene expression of Myh7, an oxidative marker, which was otherwise blunted in O mice (Appendix A).

Because abnormal mitochondrial ultrastructure is associated with impaired function, we analyzed mitochondrial morphology by TEM in the red G muscle of each group of mice. On transverse sections, each mitochondrion was manually circled within the subsarcolemmal region that was outlined in yellow (Figure 4e). The mean mitochondrion area was 20% larger in O than Y mice, but was corrected by AdipoRon (Figure 4e). In agreement with this result, the mitochondrial density (number/µm^2^) was decreased in O mice, but restored in O-AR ones (Figure 4e). Mitochondria were subsequently classified as either “healthy” (painted in green) or “damaged” (in pink). Damaged mitochondria showed at least one of the following characteristics: disarticulated cristae and vacuolization, reduced electron density of the matrix, or swollen appearance (Figure 4e inset). Mitochondria with an area above the 90th percentile of Y mice were considered swollen. The percentage of abnormal mitochondria increased considerably (by 2.4-fold) with aging, while it was normalized by AdipoRon treatment (Figure 4e). These observations indicate an increase in the size of subsarcolemmal mitochondria, a decrease in their number/µm^2^, and an accumulation of abnormal mitochondria with age. However, administration of AdipoRon counteracted these age-induced alterations.

Oxidative stress in aging muscle is intricately linked to mitochondrial dysfunction [35]. Therefore, 4-hydroxynonenal (HNE), lipid peroxidation product, and marker of oxidative stress, was measured. HNE protein levels were increased in O mice, while being significantly reduced in O-AR mice, reaching levels similar to Y ones (Figure 4f). These findings suggest that AdipoRon could help preserve mitochondrial function and reduce oxidative stress.

### 3.4. AdipoRon Attenuates Muscle Age-Related Fibrosis and NF-κB Overactivation

AMPK can be a potent suppressor of SMAD and NF-κB signaling, repressing the subsequent transcription of some pro-fibrotic and pro-inflammatory genes (Figure 5a). We thus investigated fibrosis and found that O mice displayed a stronger picrosirius staining than Y mice (+70%). By contrast, the percentage of fibrotic areas was reduced by ~20% in O-AR mice (Figure 5b,c). In addition, the active form of SMAD2 (P-SMAD2), a major downstream regulator promoting TGF-β-mediated fibrosis, was significantly higher in O mice but not in O-AR ones, where levels reached Y values (Figure 5d). Lastly, NF-κB, a pleiotropic transcription factor modulating immune and inflammatory responses [3], was studied. NF-κB activity, measured by phosphorylated p65-subunit, nearly doubled with age in O mice but returned to baseline levels with AdipoRon treatment (Figure 5e). This suggests that AdipoRon could effectively suppress NF-κB activity in aged skeletal muscle and protect it from fibrosis.

### 3.5. AdipoRon Mitigates Systemic Inflammation

Senescence-Associated Secretory Phenotype (SASP) factors including circulating cytokines and chemokines increase in the bloodstream with aging [36]. Accordingly, in this study, plasma levels of several cytokines (Figure 6a) and chemokines (Figure 6b) rose by 2- to 13-fold in O mice compared to Y ones. Except for IL-6, all factors tended to decrease or were significantly reduced by AdipoRon treatment. Some, like tumor necrosis factor α (TNF-α), interleukin-10 (IL-10), monocyte chemoattractant protein-1 (MCP-1), and macrophage inflammatory protein-2 (MIP-2), were even close to Y values (Figure 6a,b). 

## 4. Discussion

ApN is regarded as a unique and salutary adipokine exerting beneficial metabolic effects that could increase longevity. Accordingly, plasma ApN levels were higher among centenarians than in elderly individuals and these levels correlated with a favorable metabolic phenotype and a greater insulin sensitivity [37]. Correspondingly, in different pathological conditions associated with aging, decreased ApN synthesis, binding, and action led to insulin resistance and oxidative stress, while ApN treatment increased glucose uptake, mitochondrial number, and fatty acid oxidation in muscle [38,39]. Additionally, in functionally dependent geriatric patients, sarcopenia was associated with higher circulating inflammatory markers and lower ApN levels [40]. Conversely, in some life-threatening conditions associated with muscle wasting, such as chronic heart failure, ApN levels were paradoxically elevated. However, this elevation resulted from ApN resistance linked to AdipoR1 downregulation and inactivation of AMPK signaling [41]. In this situation, elevated plasma ApN reflect a protective (yet unsuccessful) mechanism to counteract ApN resistance, insulin resistance, inflammation, and stress [37,41]. In our previous study, middle aged mice displayed decreased plasma ApN [11]. Here, in much older O mice, plasma ApN levels were normalized, but there was a significant downregulation of AdipoR1 and AMPK signaling. Hence, this makes a strong case for the existence of ApN resistance in old O mice. Accordingly, these mice showed hyperglycemia possibly resulting from an age-related insulin resistance, as ApN’s insulin-sensitizing effects were hindered [2]. Importantly, a novel finding of this study was to demonstrate that administration of an AdipoR agonist is able to alleviate ApN resistance in aging, as this treatment restored AdipoR1 levels and activated the ApN signaling cascade. This is consistent with observations reported in diabetic mice [42]. As aging progresses, ApN deficiency or resistance may become prominent, potentially transitioning from one to the other. In this context, an ApN mimic could hold therapeutic promise for addressing ApN deficiency/resistance in the elderly.

An interesting study by Balasubramanian et al., conducted in old mice, showed an improved muscle function after 6 weeks of i.v. treatment with AdipoRon and a tiny increase in mitochondrial function [12]. However, because of the lack of a control group, sarcopenia could not be assessed accurately in old mice, and the effect of AdipoRon on muscle mass was not clearly established. In this study, we delve deeper into all the features of severe sarcopenia [1], namely (i) a decrease in muscle quantity as assessed by muscle weight, fiber number, and size, (ii) a decline in muscle quality as assessed by the accumulation of Tags, abnormal mitochondria, and fibrosis, and (iii) a marked reduction in muscle function as assessed by mitochondrial content/function, and endurance exercise. Additionally, we delved deeper into understanding how AdipoRon functions in sarcopenic muscle. Our findings demonstrate that three months of oral AdipoRon treatment in aged male C57/BL6J mice rescues all sarcopenia-related abnormalities and inflammaging, primarily through the AdipoR1-AMPK axis. 

In humans, a loss in muscle mass occurs incipiently from middle-age (∼1%/year) and may reach a loss of ∼50% in advanced-age. This decline is primarily due to a reduction in both muscle fiber number and the size of type 2 fibers [43]. In our 23-month-old mice, low muscle mass was assessed by reduced sarcopenia index and by morphometric analysis, conducted over whole muscle cross-sections (tibialis anterior) to avoid biases due to the selection of individual areas. This analysis showed an age-related decrease in the total number of fibers and in fiber size. A late-term treatment with AdipoRon protected mice against fiber loss due to aging, but without modifying fiber size. The lack of AdipoRon effect on fiber size is likely to be ascribed to its stimulation of oxidative metabolism (enhanced mitochondria function and increased Myh7 expression). It is known that fiber size (rather fiber type based on MyHC classification) is related to its metabolic properties [44,45] and fibers with greater oxidative metabolism are smaller than glycolytic ones [44]. Thus, a shift toward a more oxidative metabolism may prevent a re-increase in fiber size. Similarly, treatment with AdipoRon increased the number of small myofibers in dystrophic mice and improved endurance and muscle function [8].

Muscle quality was also compromised in old mice, partly due to an abundance of tubular aggregates or TAgs. TAgs are an abnormal accumulation of ordered and tightly packed sarcoplasmic reticulum (SR) tubes [46,47]. In humans, the presence of TAgs defines a clinically heterogenous group of disorders termed TA myopathies (TAM), characterized by muscle weakness [46]. Protein aggregates may also exert several adverse effects on cell function (lysosome and DNA damage, disruption of the membrane system and Ca^2+^ homeostasis, and induction of ROS), thereby accelerating aging in many ways [48]. The presence of TAgs has not yet been studied in the elderly, but virtually identical TAgs to those in TAM patients have been found in aging mice [47,49]. TAg abundance in old mice is accompanied by impaired ability to restore internal Ca^2+^ stores from extracellular space, which could contribute to muscle weakness [47]. We have previously shown that 1-year-administration of AdipoRon to young–obese mice prevented the development of TAgs [11]. Herein, we demonstrated that short-term AdipoRon given to much older mice, once TAgs have already formed [11], reversed their accumulation. This is in line with the findings observed in aged mice, where regular long-term exercise prevented the formation of TAgs and reduced muscle fatigue. [47].

The age-related decline in mitochondrial function and associated damage further deteriorates muscle quality. This impairs mitochondrial contribution to cellular bioenergetics, increases ROS production, and may induce mitochondrial membrane permeabilization, leading to inflammation, death, and accelerated sarcopenia [35]. Accordingly, we observed a decrease in COX activity in muscles of O mice, indicating impaired oxidative capacity and mitochondrial function. Ultrastructural analysis of electron microscopy images revealed swollen and damaged mitochondria with age. Oxidative stress, reflected by HNE, was also increased. All these abnormalities were completely corrected by AdipoRon. 

Since AdipoRon decreased TAgs accumulation and improved mitochondrial health, we explored its potential to enhance pathways responsible for clearing misfolded proteins. As reported in some studies [50], the ubiquitin–proteasome system was downregulated in sarcopenia and remained unmodified by AdipoRon. Moreover, this system is helpless when facing large misfolded proteins such as protein aggregates or damaged organelles [48]. We thus examined the effects of AdipoRon on the autophagy–lysosome system. Different lines of evidence indicate that autophagy declines with aging, while autophagy stimulation may have beneficial effects on healthspan and lifespan [28,51]. Autophagy was impaired in our old mice, shown by p62 protein accumulation. In contrast, AdipoRon effectively boosted autophagy, as evidenced by an increased LC3-II/LC3-I ratio and inhibited p62 accumulation. This was explained by upstream activation of AMPK, which in turn activates ULK1 and Beclin-1, two kinases that serve as initiators of autophagy [28]. Autophagy is not only involved in proteostasis, but also affects entire organelles (including dysfunctional mitochondria targeted by ‘‘mitophagy,’’ a selective type of autophagy) [28,35]. Mitophagy is also known to be impaired in aged skeletal muscle, leading to progressive accumulation of defective mitochondria [52]. In AdipoRon-treated mice, the presence of damaged/dysfunctional mitochondria was erased, indicating improved clearance of defective organelles. The higher mitochondrial content (see TOMM20) observed in treated mice, despite higher clearance, further suggests that AdipoRon could enhance mitochondrial biogenesis via PGC-1α. Taken together, our findings suggest that AdipoRon protects muscle by reducing age-induced TAgs and damaged mitochondria accumulation through autophagy/mitophagy stimulation and potentially by promoting mitochondrial biogenesis.

Fibrosis also significantly impacts muscle quality in the elderly by compromising structural integrity, function, and regenerative processes. It is a key factor contributing to muscle weakness, stiffness, and vulnerability to reinjury [53]. O mice displayed fibrosis, as shown by marked collagen deposits and high levels of P-SMAD2, a TGF-β effector that regulates fibrosis. Under AdipoRon, collagen accretion was attenuated and P-SMAD2 levels were normalized. AMPK can be a powerful suppressor of TGF-β/SMAD signaling [54]. Conversely, specific AMPK inhibition in mouse fibro-adipogenic progenitors has been shown to enhance TGF-β signaling and promote fibrosis in regenerated muscles [55]. This is explained by the potent repressive role of AMPK on NF-κB through the Sirtuin-1/PGC-1α pathway [3]. So, impaired AMPK results in enhanced NF-κB activity and subsequent transcription of pro-inflammatory and pro-fibrotic genes [55,56]. Herein, O mice exhibited a ~2 fold-rise in NF-κB activity, to which mitochondrial dysfunction, oxidative stress, and inflammation are likely to contribute [57]. Such a rise was completely abolished by AdipoRon treatment. Taken together, our data indicate that AdipoRon could be a therapeutic prospect to dampen fibrosis in aged skeletal muscle.

Aging brings about inflammaging, a chronic low-grade systemic inflammation, which is associated with various age-related diseases including cardiovascular disease, cancer, type 2 diabetes, cognitive decline, and sarcopenia/frailty [58]. Over time, persistent inflammation may render muscles susceptible to sarcopenia by promoting protein degradation, hindering regeneration, disrupting mitochondrial function, and inducing insulin resistance [36]. Inflammaging prevailed in O mice but was mitigated by AdipoRon. Most cytokines and chemokines were (or tended to be) decreased by treatment, with a striking normalization for some (TNF-α, IL-10, MCP-1, or MIP-2). The surge of the anti-inflammatory cytokine IL-10 in O mice may be viewed as a compensatory mechanism to counterbalance systemic inflammation [49]. With regard to TNF-α, high circulating levels were associated with an 8-fold increased risk of sarcopenia in elderly subjects [59], while its pharmacological blockade prevented sarcopenia and extended survival in aged mice [60]. Moreover, circulating MCP-1 has been described as a measure of biological rather than chronological age both in mice and in humans. In mice, MCP-1 levels were markedly elevated in progeroid animals, while in humans, they were higher in elderly frail individuals compared to non-frail individuals [61]. Our findings show that AdipoRon effectively dampens inflammaging, a root cause of numerous chronic diseases in the elderly.

AdipoRon’s beneficial and protective effects culminated in enhanced muscle function, demonstrated by increased endurance in treadmill tests. Our results build upon previous findings, indicating improved agility in vivo, and greater resistance to fatigue and quicker recovery in glycolytic muscle ex vivo after 6 weeks of AdipoRon treatment in old mice [12]. Here, the endurance boost is due to significantly improved mitochondrial function. It could also be attributed to reduced dysfunction in SR proteins and the subsequent improved capability of fibers to use extracellular Ca^2+^ [47]. Reduced fibrosis, oxidative stress, and inflammaging might contribute as well. Overall, these findings align with AdipoRon’s exercise-mimicking properties [2,6].

In this study, mice were studied in a basal state to directly assess the effects of AdipoRon on age-related muscle changes. However, in these conditions, certain processes are weak or some cell types are scarce and need to be unveiled by muscle challenge (i.e., injury) [62]. Therefore, consistent with prior reports [62], protein levels of p16 and p21, markers of senescent cells, were undetectable. Additionally, we only found a slight re-increase in Myogenin protein levels under AdipoRon. Muscle injury typically increases senescent cell numbers, which, via their released SASPs, trigger inflammation and fibrosis, preventing muscle regeneration [62]. Therefore, exploring AdipoRon’s potential impact on injured muscles in aged mice would be valuable. In addition, investigating AdipoRon’s effects on senescence in DMD, a disease marked by cycles of necrosis, repair, fibrosis, and weakness [2,3] could yield new insights. We have previously shown AdipoRon’s beneficial impact on the dystrophic phenotype [8] and exploring its role effects in senescence might shed light on the underlying mechanisms. Furthermore, untreated old mice appeared to have a higher tumor prevalence. Given the anticancer properties of ApN and AdipoRon [2], due to their effects on reducing inflammation, stress, and cell proliferation [63] and given that long-term treatment with AdipoRon is well-tolerated [11], it would be crucial to investigate the potential long-term protection of AdipoRon against age-related cancer. Ideally, larger cohorts of old mice would be treated, possibly for an extended duration. Eventually, examining effects on survival curves and lifespan could provide further insights. Finally, expanding the study onto different strains and sex of mice would also be interesting.

## 5. Conclusions

Regular exercise is a powerful anti-aging strategy that can delay morbidity, extend lifespan, and promote longevity, mainly by activating the AMPK-PGC-1α pathway, improving mitochondrial function, and enhancing muscular endurance. However, implementing exercise, especially in the elderly, can be a challenging or even an impossible task. Therefore, efforts to find drugs that target the same pathways have become a hot topic in the field. In this study, we clearly demonstrate that a late three-month administration of AdipoRon in aged male C57/BL6J mice yielded notable improvements in muscle mass, quality, and function. AdipoRon treatment effectively corrected all the abnormalities associated with sarcopenia, mainly via the AdipoR1-AMPK axis. These findings, coupled with those of our previous study, demonstrate the remarkable potential of AdipoRon as a therapeutic intervention for addressing age-related muscle decline and improving healthspan.

## Figures and Tables

**Figure 1 antioxidants-13-01073-f001:**
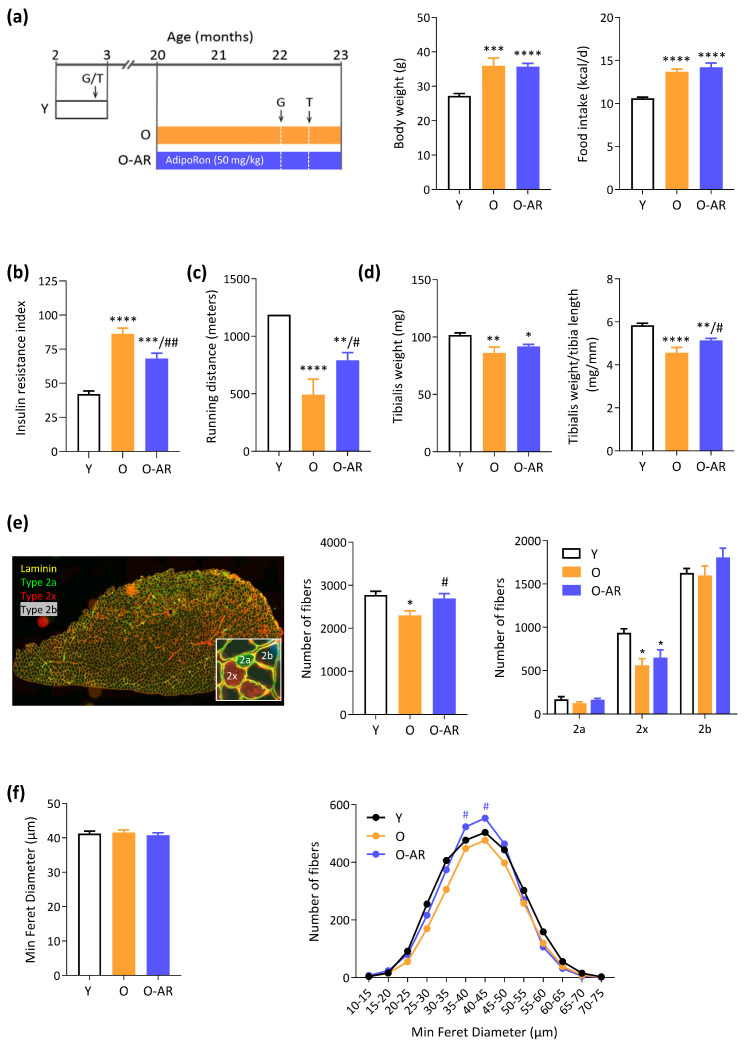
AdipoRon reduces insulin resistance, improves endurance, and attenuates sarcopenia in O mice. (**a**) Experimental protocol. Three groups of mice were studied. Two groups were studied from 20 months to 23 months of age and are referred to as old (O) mice. One of the two groups was orally treated with AdipoRon (AR) (50 mg/kg/day; O-AR), while the other one was left untreated (O). An additional group of young (3-months-old) mice was also used for comparison. At the indicated times, fasting glycemia was measured (G) and mice were submitted to treadmill exhaustion test (T). Body weight at the end of the protocol and average food consumption over the whole study. (**b**) Insulin resistance index calculated as Insulin (ng/mL) × Glycemia (mg/dL). (**c**) Running distance covered (m) in an uphill treadmill exhaustion test. (**d**) Tibialis anterior weight and the sarcopenia index, calculated as TA muscle weight normalized to tibia length. (**e**) Fiber typing was carried out by immunofluorescence staining of different MyHCs on a transversal cross-section of TA muscle. Type 1 fibers were labeled in blue (none), type 2a in green, and 2x in red while 2b were nonlabeled (black). A laminin antibody was used to delineate the basal membrane (yellow). Total number of fibers and number of each fiber type. (**f**) Global fiber size based on the Minimum Feret Diameter and fiber size distribution. All these data were quantified on whole cross-sections of TA. Data are means ± SEM for 6 Y, 6 O, and 6–9 O-AR. Statistical analysis was performed using one-way ANOVA followed by Tukey’s test to compare the three groups of mice. * *p* < 0.05, ** *p* < 0.01, *** *p* < 0.001, **** *p* < 0.0001 vs. Y mice. # *p* < 0.05, ## *p* < 0.01 vs. O mice.

**Figure 2 antioxidants-13-01073-f002:**
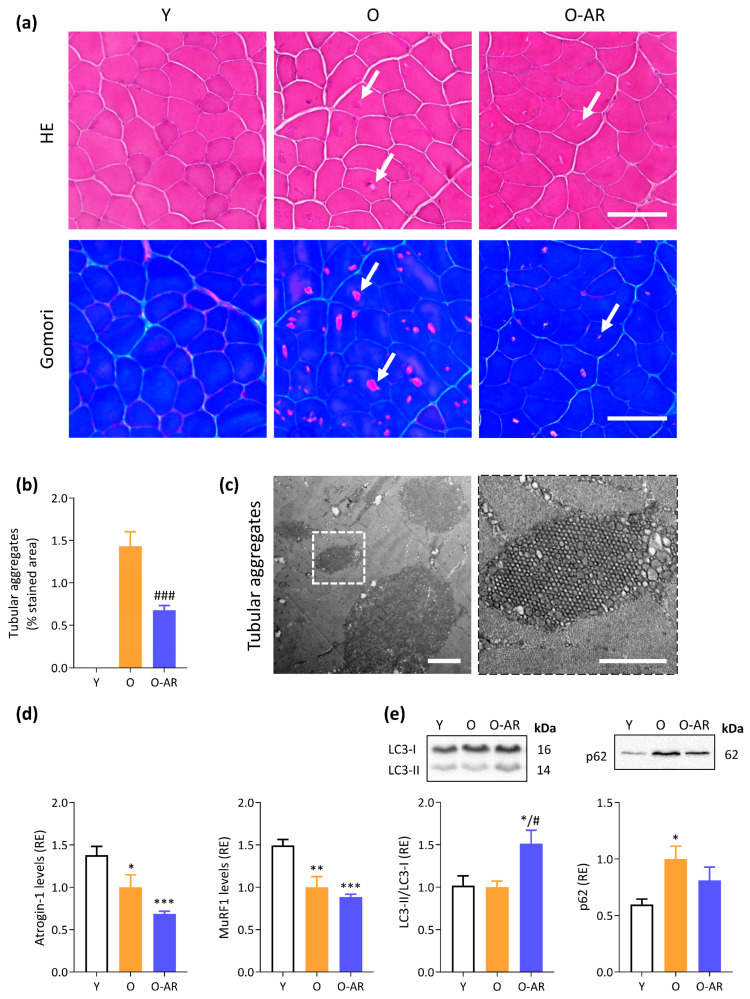
AdipoRon reduces age-related accumulation of tubular aggregates (TAgs). (**a**) TAgs (indicated by arrows) appear as pale or slightly basophilic inclusions with hematoxylin–eosin (HE) and bright pink ones with Gomori Trichrome on serial transversal cross-sections of TA from the three groups of mice (Y, O, and O-AR). Scale bars = 100 µm. (**b**) Quantification of TAg abundance after Gomori Trichrome was expressed as a % of stained area (i.e., bright pink inclusions areas normalized to the cross-sectional area of the muscle). (**c**) Characteristic honeycomb appearance of TAgs by transmission electron microscopy. Scale bar = 2 µm, Inset: higher magnification of a honeycomb TAg (scale bar = 1 µm). (**d**) Protein levels of Atrogin-1 and MURF1 were measured by ELISA. (**e**) LC3-II/LC3-I ratio and p62 were analyzed by Western blotting. p62 levels were normalized to Ponceau S staining (shown in Appendix A). Results are presented as relative expression compared to O values (**d,e**). Data are means ± SEM for 6 Y, 6 O, and 9 O-AR. Statistical analysis was performed using unpaired two-tailed *t*-test to compare O and O-AR (**b**) or one-way ANOVA followed by Tukey’s test to compare the 3 groups of mice (**d,e**). * *p* < 0.05, ** *p* < 0.01, *** *p* < 0.001 vs. Y mice. # *p* < 0.05, ### *p* < 0.001 vs. O mice.

**Figure 3 antioxidants-13-01073-f003:**
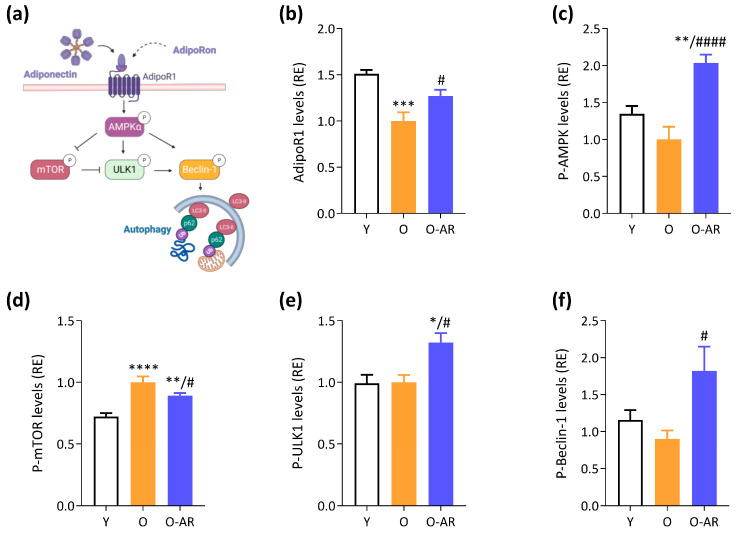
AdipoRon activates the AMPK pathway and promotes autophagy. (**a**) Proposed model for the effects of AdipoRon. Briefly, AdipoRon binds to AdipoR1 and activates the AMPK. AMPK plays a pivotal role in initiating autophagy by activating ULK1 and Beclin1, two critical regulators of the autophagic process. AMPK also exerts its regulatory influence by inhibiting mTOR, thereby alleviating mTOR-mediated inhibition of ULK1. LC3-II and p62 are two important proteins in the autophagy process. Pointed arrows indicate activation, while blunt arrows indicate inhibition. (**b**) AdipoR1 protein levels were measured by ELISA. Phosphorylated and active forms of (**c**) AMPKα (P-AMPK), (**d**) mTOR (P-mTOR), (**e**) ULK1 (P-ULK1), and (**f**) Beclin-1 (P-Beclin-1) were quantified by ELISA. Absorbance data were presented as relative expression compared with O values. Data are means ± SEM for 6 Y, 6 O, and 9 O-AR. Statistical analysis was performed using one-way ANOVA followed by Tukey’s test to compare the 3 groups of mice. * *p* < 0.05, ** *p* < 0.01, *** *p* < 0.001, **** *p* < 0.0001 vs. Y mice. # *p* < 0.05, #### *p* < 0.0001 vs. O mice.

**Figure 4 antioxidants-13-01073-f004:**
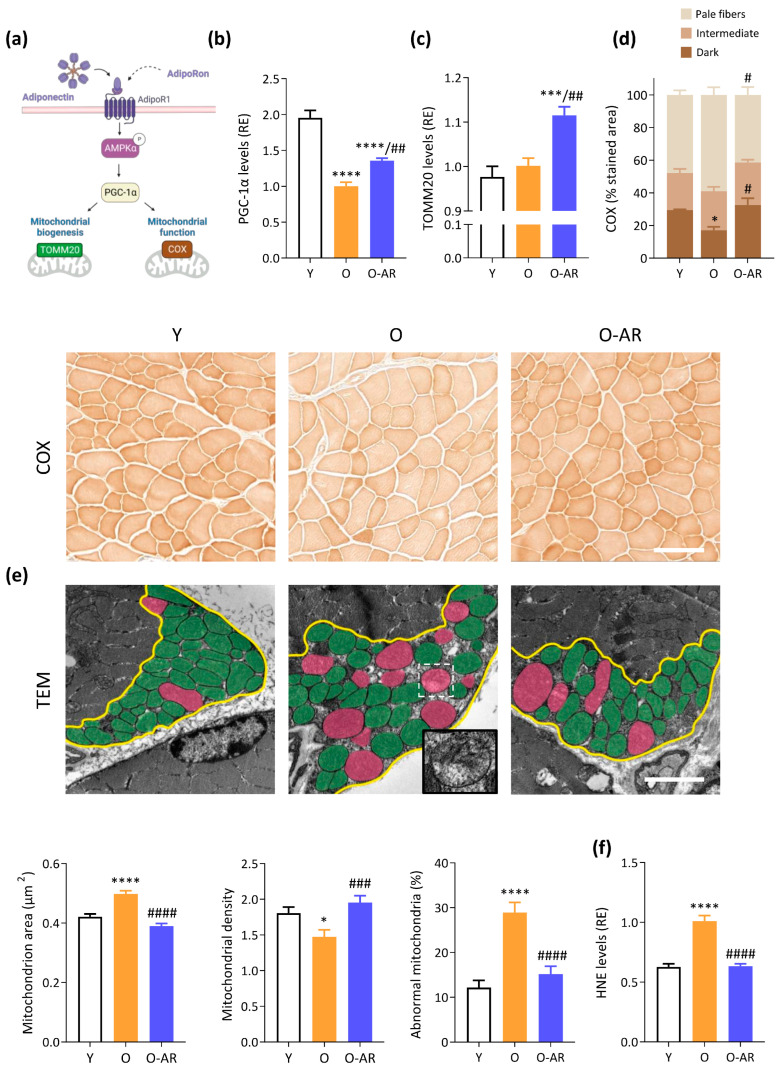
AdipoRon enhances mitochondrial biogenesis and function. (**a**) Activation of the AMPK-PGC-1α signaling pathway by ApN stimulate mitochondrial biogenesis and function. TOMM20 is a marker of mitochondrial content, while COX activity reflects mitochondrial activity. Pointed arrows indicate activation or induction. (**b**) PGC-1α and (**c**) TOMM20 protein levels quantified by ELISA. (**d**) Quantification of COX activity in the TA from the three groups of mice (Y, O and O-AR) based on histochemistry staining, for which representative transversal cross-sectional images are shown after. Three staining intensities were defined as pale, intermediate, or dark, with the darkest color being associated with the highest activity. Activity was expressed as % of stained area (i.e., colored areas normalized to the cross-sectional area of the muscle). Scale bar = 100 µm. (**e**) Transmission electron micrographs (TEM) of mitochondria in the subsarcolemmal region of G (transversal cross-sections) in the 3 groups of mice (Y, O, and O-AR). The yellow line delimits the subsarcolemmal area. Abnormal mitochondria are false-colored in pink, while normal mitochondria are colored in green. Representative images of each group are shown. Scale bar = 2 µm. The mitochondrion area and the number of mitochondria per µm^2^ in the subsarcolemmal region were calculated based on TEM images. The percentage of abnormal mitochondria out of the total number of mitochondria was also calculated. In total, 1117, 1604, and 1435 subsarcolemmal mitochondria were sampled for Y, O, and O-AR group, respectively. (**f**) HNE protein levels were quantified by ELISA. (**b,c,f**) Absorbance data were presented as relative expression compared with O values. Data are means ± SEM for 6 Y, 6 O, and 9 O-AR (**b**–**d**,**f**). Statistical analysis was performed using one-way ANOVA followed by Tukey’s test to compare the 3 groups of mice. * *p* < 0.05, *** *p* < 0.001, **** *p* < 0.0001 vs. Y mice. # *p* < 0.05, ## *p* < 0.01, ### *p* < 0.001, #### *p* < 0.0001 vs. O mice.

**Figure 5 antioxidants-13-01073-f005:**
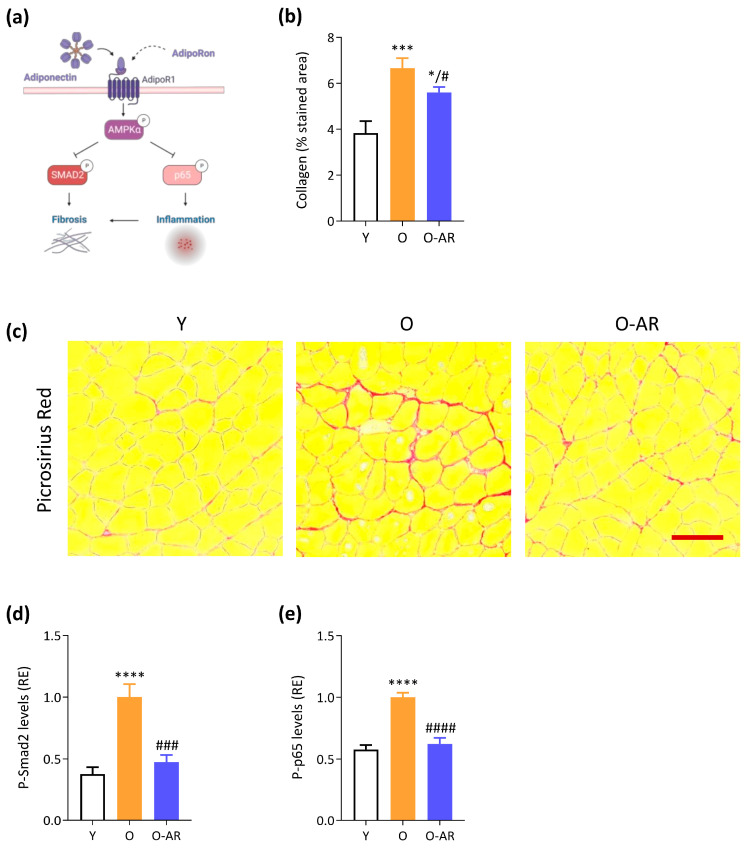
AdipoRon limits age-related fibrosis and inflammation**.** (**a**) Activation of the AMPK signaling pathway by ApN represses SMAD2 activity, an effector protein of the TGF-β pathway, and NF-κB activity (p65 subunit). Pointed arrows indicate activation, while blunt arrows indicate inhibition. (**b**) Quantification of picrosirius red expressed as a % of the collagen stained area, for which representative transversal cross-sectional images are shown in (**c**), scale bar = 100 µm. The active, phosphorylated forms of (**d**) SMAD2 (P-SMAD2) and (**e**) p65-subunit of NF-κB (P-p65) were measured by ELISA. Absorbance data were presented as relative expression compared with O values. Data are means ± SEM for 6 Y, 6 O, and 9 O-AR. Statistical analysis was performed using one-way ANOVA followed by Tukey’s test to compare the 3 groups of mice. * *p* < 0.05, *** *p* < 0.001, **** *p* < 0.0001 vs. Y mice. # *p* < 0.05, ### *p* < 0.001, #### *p* < 0.0001 vs. O mice.

**Figure 6 antioxidants-13-01073-f006:**
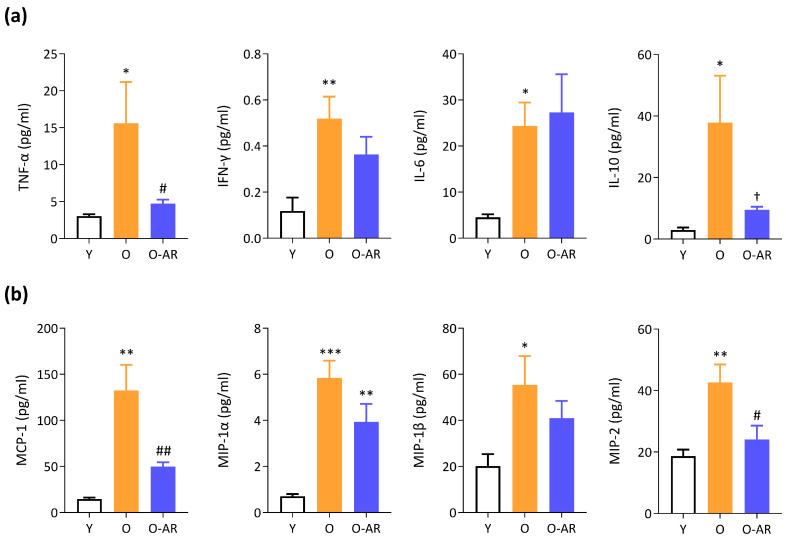
AdipoRon partially protects against age-related systemic inflammation. (**a**,**b**) A multiplex assay was performed to measure the concentrations of different cytokines (**a**) and chemokines (**b**) in the plasma of the three groups of mice. These can serve as markers of inflammaging/senescence associated secretory phenotype (SASP). Data are means ± SEM for 6 Y, 6 O, and 9 O-AR. Statistical analysis was performed using one-way ANOVA followed by Tukey’s test to compare the 3 groups of mice. * *p* < 0.05, ** *p* < 0.01, *** *p* < 0.001 vs. Y mice. † *p* = 0.07, # *p* < 0.05, ## *p* < 0.01 vs. O mice.

## Data Availability

All of the data is contained within the article and the Appendix A.

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
