# Peer review of "Challenging Sarcopenia: Exploring AdipoRon in Aging Skeletal Muscle as a Healthspan-Extending Shield"

_antioxidants, 2024, doi:10.3390/antiox13091073_

Round 1

Reviewer 1 Report

The aim of this study was to evaluate the potential of the adiponectin receptor agonist AdipoRon as a treatment for age-related sarcopenia in C57BL/6J mice. Overall, the data suggest that this compound can ameliorate the effect of age on sarcopenia in mice due to differential autophagic activity, mitochondrial function and pro-inflammatory cytokine signaling. The experimental design is adequate, although only male mice were evaluated. Interestingly, there was no effect on the circulating adiponectin level in AdipoRon treated mice which suggests there could be off target effects of this compound on muscle function. That is, the compound is having an effect in a non-adiponectin receptor dependent manner.    

Specific comments:

What was the sample size for the young mice?

How was the dose of AdipoRon chosen? Is it possible to quantify the circulating AdipoRon level in the treated mice?

Is a single training session sufficient for the treadmill testing given that older mice will presumably learn the task more slowly than young mice?

How was the insulin resistance index generated if circulating insulin level was not measured in the mice?  There is no mention of it being measured in the materials, nor are there insulin data reported. The circulating glucose data are also not reported.

There are no loading controls provided for the Western blot data.

Is it known if mice with defects in autophagic function have a higher prevalence of tubular aggregates? If there was no effect of AdipoRon in these models, this would provide more compelling evidence for a direct link.

Why wasn’t mitochondrial number quantified using a more direct measure such as Mitotracker?

Page 15, line 513.  To say these data unequivocally demonstrate an effect of AdipoRon is too strong.  It clearly has an effect in this specific strain, and this sex.  There are many instances where a treatment/intervention has an effect in once strain of mice, but not another, due the differences in genetic backgrounds.

Page 18, Line 631. See comment above.  These data strongly support the notion that AdipoRon is beneficial in this context in this model, but it is certainly not conclusive.

Although there is the anecdotal observation that the tumor incidence was higher in the untreated old mice, this is based in a very small sample size.  It is premature to suggest that AdipoRon has an anti-tumorigenic effect.  

Author Response

We thank the reviewer for taking the time to review our manuscript, and for the valuable comments and remarks. We have provided a point-by-point response to all comments in the attached document. 

Reviewer 2 Report

Review of the manuscript titled "Challenging Sarcopenia: Exploring AdipoRon in Aging Skeletal Muscle as a Healthspan-Extending Shield,"

Suggested improvements:

1. The introduction section should explicitly state the research hypothesis and objectives more clearly. Currently, the aim is mentioned broadly: "This study aimed to determine if AdipoRon treatment could effectively reverse sarcopenia and hallmarks of aging in older mice." (Lines 71-72).

2. Include more recent studies related to AdipoRon and sarcopenia to provide a comprehensive background and justify the novelty of the research. "AdipoRon, an orally active synthetic agonist of ApN receptors, has shown promise in animal models..." (Lines 54-56).

3. Provide more details about the selection criteria for the mice, the environmental conditions, and any exclusion criteria for the study subjects. "20-month-old male C57BL/6J mice (Jackson Laboratory, Bar Harbor, USA) were divided into two groups..." (Lines 79-80).

1. The manuscript should explain the rationale for including the specific control groups and how they were managed. "Young 3-month-old mice (Y) were used as controls for all parameters studied." (Lines 241-242)

2. Provide deeper insights into the mechanistic pathways through which AdipoRon exerts its effects, with more emphasis on molecular signaling pathways. "To understand how AdipoRon might boost autophagy, we examined receptor protein levels and the activity of downstream kinases..." (Lines 313-314)

3. Enhance the discussion by comparing findings with existing literature, discussing potential implications, and acknowledging any limitations. "Herein, we demonstrated that short-term AdipoRon given to much older mice, once TAgs have already formed, reversed their accumulation." (Lines 540-541)

4. Discuss the higher incidence of tumors in untreated old mice and its potential impact on study results and conclusions. "During the study, 3 out of 9 mice in the untreated O group developed large intra-abdominal tumors..." (Lines 238-239)

5. Propose specific future research directions based on the findings of this study, addressing any gaps or unanswered questions. "Therefore, exploring AdipoRon's potential impact on injured muscles in aged mice would be valuable." (Lines 623-624)

6. Discuss the potential therapeutic applications of AdipoRon in human clinical settings more extensively. "Conclusively, a three-month administration of AdipoRon in aged mice yielded in notable improvements in muscle mass..." (Lines 631-632)

Author Response

(The authors gave the same response as above.)

Round 2

Reviewer 1 Report

The authors have adequately addressed each of my comments.  I have no further suggestions. 

NA

Author Response

We greatly appreciate your valuable comments and suggestions, which have significantly contributed to enhancing the quality of our manuscript

Reviewer 2 Report

The responses to my previous comments are satisfactory and I do not have other comments, so I conclude for acceptance in the present form.

-

Author Response

(The authors gave the same response as above.)
